# *Broussonetia papyrifera* Extract Can Be Used as a Raw Material Source for a Sterility Agent for *Microtus fortis*

**DOI:** 10.3390/biology14010056

**Published:** 2025-01-12

**Authors:** Shuangye Wang, Tian Lan, Yunlin Zhao, Wanfu Liu, Tian Huang, Meiwen Zhang, Zhiyuan Hu, Zhenggang Xu

**Affiliations:** 1The Central Laboratory of Medical Molecular Biology, School of Basic Medicine, Guiyang Healthcare Vocational University, Guiyang 550081, China; 18874772324@163.com; 2Hunan Engineering Research Center of Ecological Environment Intelligent Monitoring and Disaster Prevention and Mitigation Technology in Dongting Lake Region, College of Information and Electronic Engineering, Hunan City University, Yiyang 413000, China; huangtian@hncu.edu.cn (T.H.); huzhiyuan2019@163.com (Z.H.); 3Hunan Research Center of Engineering Technology for Utilization of Environmental and Resources Plant, Central South University of Forestry and Technology, Changsha 410004, China; rssq198677@163.com (Y.Z.); 20221100208@csuft.edu.cn (W.L.); 4Key Laboratory of National Forestry and Grassland Administration on Management of Western Forest Bio-Disaster, College of Forestry, Northwest A & F University, Yangling 712100, China; 3389769062@nwafu.edu.cn; 5Dongting Lake Station for Wetland Ecosystem Research, Institute of Subtropical Agriculture, The Chinese Academy of Sciences, Changsha 410125, China; zhangmw@isa.ac.cn

**Keywords:** methanolic extract, *Microtus fortis*, paper mulberry, sterilants, organ coefficient, sperm teratogenicity

## Abstract

Rodent pests frequently cause damage to crops; *Microtus fortis* mainly causes agricultural losses in the Dongting Lake wetland, China. In recent years, the use of antifertility substances from certain plants instead of lethal rodenticides has been considered adequate to avoid negative effects on ecosystems. *Broussonetia papyrifera* grows fast and is widely distributed, including in the Dongting Lake wetland. When *M. fortis* migrates, it feeds on the leaves of *B. papyrifera*, which it finds palatable; the basis for a personalized sterility agent for *M. fortis* can be established using these leaves. According to this study, the different contents of *B. papyrifera* leaf ethanol extracts had an inhibitory effect on the reproductive organs of *M. fortis*. Notably, they had a more significant inhibitory effect on *M. fortis* males, which included reducing their testosterone levels and sperm ability, and controlled the offspring parameter. These findings enrich the study of plant sterilants and provide insights into the utilization of *B. papyrifera* for the management of rodents.

## 1. Introduction

Rodents are among the most species-rich mammals. They comprise approximately 42% of all living mammals, with over 2200 defined species [1] including various kinds of mice, rats, and voles. Some are significant pests associated with crop damage and the spread of disease globally [2,3,4]. Among vertebrates, rodents are primarily responsible for damage to forests and agriculture [5,6,7]. Methods have been developed to control rodent pests, enriching the literature on rodent population biology [8,9,10]. In recent years, chemical methods have become the mainstay of controlling rodent pests. These include anticoagulant rodenticides, which can inhibit vitamin K epoxide reductase, an important enzyme in the production of blood clotting factors [11].Exposed animals typically have a prolonged blood clotting time, which leads to organ function interference, which has lethal consequences [12]. However, rodents are mammals with sophisticated behavior, and it is essential to consider behavioral responses that have evolved via natural selection to increase individual fitness [13,14], enabling them to resist these chemical rodenticides over time. The first detection of a resistant rodent strain was reported in Scotland in 1958, and similar reports followed from other areas in Europe [15]. To solve this problem, newer rodenticides were developed, though they were challenged by cross-resistance to previous rodenticides [16]. These resistances may be related to chromosomes inherited by successive generations of rodents [17]. Though lethal chemical rodenticides may deal with an immediate pest problem by causing rodent mortality, lethal control may be followed by the rapid immigration of rodents so that the reduction in damage is short-lived [18]. This consequence not only has led to potential harm to rodent predators and the environment [19,20] but might also result in rodent resistance to the toxicant used, as some of them ingest small doses and generate immunologic memory [21].

Recent attempts to control rodent populations have included fertility management, which is a hot topic in pest prevention. These studies have advocated for sterilization to control fertility [22,23,24]. Conventional sterilization methods include immunotherapy, surgical sterilization, and gene drives [23,25,26]. For example, immunocontraception uses an individual’s immune system to inhibit fertility [27], whereas chemosterilization uses a chemical or hormonal compound [28]. Though they have the potential to be effective based on their potency, rodent resistance to these sterilization methods has evolved for the same reasons as resistance to rodenticides [25,29]; additionally, they are costly and may impose ecological risks [30,31]. Plant-derived sterilants are mainly produced from plants and their extracts, have more persistent control effects compared with conventional chemical agents, and are more environmentally friendly and economic [17]. At present, abundant plant species have been known to inhibit the reproduction of animals [32]. Among them, triptolide and curcumol are used most often in rodent control. Triptolide is extracted from *Tripterygium wilfordii*, and its antifertility effect on male rats intensifies with higher doses [33]. Curcumol is an important component of the essential oil of *Rhizoma curcuma* and has antifertility effects on both sexes of rodents [34].

*Microtus fortis* is a small herbivore species widely distributed around Eurasia; this species of vole often damages agriculture in countries like China, Russia, Mongolia, and Korea [35]. Meanwhile, the vole is the host of some pathogens like hantavirus, which causes hemorrhagic fever with renal syndrome in human [36]. Because of the species’ rapid reproduction, *M. fortis* is the dominant rodent around the Dongting Lake region, in the middle of the Yangtze valley, China [37,38]. During the dry season, these voles primarily live on the beaches of the Dongting Lake wetland. However, the available space decreases during the wet season, and its population density increases; the voles climb over the flood bank of the Dongting Lake wetland and migrate to the surrounding villages and cropland [39], where they damage crops and trigger health events. To prevent the voles from migrating outside of the wetland, rodent-proof walls have been built around the Dongting Lake wetland [40]. When the vole population breaks out, people usually kill them by leveraging artificial behaviors or using chemical rodenticides [41]. These methods are effective but lead to environmental problems. For example, highly toxic rodenticides like anticoagulants are effective for controlling rodents but also cause the death of their predators and other species due to broad-spectrum toxicants [42,43]. Therefore, developing some affiliative inhibitors such as plant-based inhibitors is more friendly to the ecosystem.

*Broussonetia papyrifera* is a deciduous trees belong to the Moraceae family; these trees grow fast with high adaptability and are widely distributed around the Dongting Lake region [44]. In the wet season, *M. fortis* feeds on *B. papyrifera* leaves during its migration from the beaches to the croplands. Owing to their good palatability, *B. papyrifera* leaves have been used to feed livestock for a long time [45,46]. However, some compounds in *B. papyrifera* leaves inhibit estrogen biosynthesis in mammalian ovarian granulosa [47]. Previous feeding experiments revealed that the growth and fertility of *M. fortis* were inhibited after being fed *B. papyrifera* leaves [48]. A subsequent study found that *M. fortis* is strongly attracted to *B. papyrifera* leaves for feeding; meanwhile, substances in *B. papyrifera* leaves significantly transform in response to vole bites [49]. Given that substances in *B. papyrifera* leaves are abundant and applicable in the environment, the leaves have been suggested as a potential resource for developing plant-based inhibitors used to control the reproduction of rodents. The present study aimed to explore the antifertility activity of *B. papyrifera* leaf methanol extract on the reproduction of *M. fortis* in the laboratory. Moreover, as *M. fortis* is a suitable animal model, this study may provide a scientific basis for comprehensive rodent control and management.

## 2. Materials and Methods

### 2.1. Plant Material

Fresh *B. papyrifera* leaves were collected from the campus of Central South University of Forestry and Technology in Changsha, China (28°6′25.48″ N, 112°59′37.68″ E). The leaves were first thoroughly washed using ultrapure water, air-dried in the shade with optimal ventilation, and finally placed in a constant-temperature oven at 60 °C for accelerated drying. The dried leaves were ground into coarse powder and then macerated in a tenfold (*w*/*v*) volume of 100% methanol for 3 days at room temperature. The residue was filtered using a screen (0.075 mm). Methanol was vaporized under reduced pressure to acquire the *B. papyrifera* leaf methanol extract. The extract was 10-fold, 100-fold, 500-fold, 1000-fold, and 5000-fold dissolved with ultrapure water, and five doses of the extract were obtained for each dissolution.

### 2.2. Animals and Experimental Design

A total of 90 8-week-old *M. fortis* males and 90 8-week-old *M. fortis* females were used in this study. They were the offspring of wild-caught *M. fortis* from the Dongting Lake wetland that was maintained in the laboratory as outbred stock. The voles were assigned into six groups, including one control group and five treatment groups. Each group contained 15 males and 15 females. Each *M. fortis* was maintained under standard conditions with 12 h light/dark cycles at 20–22 °C, and water and fodder (Hunan SJA Laboratory Animal CO., LTD., Changsha, China) were freely available [50]. The voles were weighed before the experiment to record their original weight. The voles in the control group (CK) were administered an additional 1 mL of ultrapure water per day via gavage. The voles in the five treatment groups were administered an additional 1 mL of the 5000-fold (group I), 1000-fold (group II), 500-fold (group III), 100-fold (group IV), and 10-fold (group V) dissolved extract per day (Figure 1).

### 2.3. Anatomy and Morphological Index of Organs

For both males and females in each group, 12 individuals were separated into three cages. Voles in one of the three cages were treated for 1 day. They were then sacrificed on the second day to collect data and samples. The voles in the other two cages were treated for ten and twenty days, respectively; they were all sacrificed on the second day after their last treatment. The voles were weighed and then excessively anesthetized with diethyl ether [51] after the last gavage for 24 h. The heart, liver, lungs, kidneys, testes, uterus, and ovaries were excised, washed, and weighed. Moreover, the length of the uterus was measured, and the volume of the testes was estimated according to the following equation:V = 4π/3(long diameter/2 × transverse diameter/2 × anteroposterior diameter/2)(1)

### 2.4. Hormone Assessment

Blood samples were immediately collected from narcotized voles by removing their eyeballs [52]. The blood samples were kept at room temperature for 1 h and then were centrifuged twice at 5000 rpm for 10 min to obtain serum, which was transferred into new tubes and stored at −20 °C for subsequent measurements. The males’ serum testosterone and luteinizing hormone (LH) and the females’ estradiol, follicle-stimulating hormone (FSH), and LH were quantified using enzyme-linked immunosorbent assay (ELISA) kits purchased from Nanjing Jiancheng Bioengineering Institute (Nanjing, China), following the manufacturer’s instructions.

### 2.5. Sperm Analysis

The left cauda epididymidis was carefully separated from each testis after washing with PBS. The cauda epididymis was placed in 2 mL of normal saline with 10% BSA, cut, and incubated at 37 °C for 15 min. The sperm suspension was analyzed using a computer-aided sperm analysis system (CASA, Song Jing Tian Lun Biotechnology Co., Ltd., Nanning, China). More than 5 sights of each vole were captured, and sperm quantity and vitality were examined and recorded. For the sperm morphology assessment, a smear of sperm sample was fixed and dyed using the Diff-Quik staining method [53]. The slides were observed under a light microscope under 40× magnification; at least 20 sights of each vole were inspected randomly for the recording of the abnormal and total quantity of sperm.

### 2.6. Mating and Breeding Assay

Three *M. fortis* males and females remained in each group for breeding. After receiving treatment for 20 days, these males and females were randomly matched and placed in three cages, and treatment was stopped. The reproduction situation in each cage was inspected each day. When fetal voles appeared, the time of reproduction and the number and weight of the fetuses were recorded.

### 2.7. Data Statistics

The organ coefficients were calculated using the weights of the organs divided by the weights of the bodies. The mean values were calculated using SPSS 19.0 software and expressed as means ± standard error (Mean ± SE). Significant differences among the mean weights of the cages were analyzed using one-way analysis of variance (ANOVA), and weight variations in the cages throughout the experiment were analyzed using the paired samples *T* test method. Concerning reproduction, this study compared the results of the CK group with those of each treatment group, and the independent samples *T* test was used to evaluate the differences. A *p*-value lower than 0.05 (*p* < 0.05) was considered statistically significant.

## 3. Results

### 3.1. Variation In Body Weight Response to Extract

To confirm the influence of the *B. papyrifera* leaf extract on the weight of these voles, different groups were compared. The results show that the weights changed fractionally during treatment (Figure 2). The weight range of the CK was stable, with discrepancies lower than 2%, and the weights of the treated groups were at a similar level to that of the CK. The slight variations are possibly related to food intake and digestion. Therefore, the *B. papyrifera* leaf extract showed a marginal effect on the body weight of *M. fortis*.

### 3.2. Influence of Extract on the Organ Coefficient

Overall, there was no obvious influence of the *B. papyrifera* leaf extract on the development of the hearts, livers, lungs, and kidneys of *M. fortis*. The levels of these organ coefficients of male voles were similar (Figure A1A). A few significant differences were found between some of the treated groups and the CK (*p* < 0.05), though most organ coefficients showed no significant difference during the experiment. Fewer differences were found in females compared to in males (Figure A1B); there was no significant difference in the lung coefficients or liver coefficients between the treated groups and the CK (*p* > 0.05). Only two significant differences in the heart coefficients and one in the kidney coefficients were found by comparing the treatment and CK voles (*p* < 0.05).

Figure 3 summarizes the comparison of sex organs of *M. fortis* between each treated group and the CK. As a whole, the inhibitory effect of the *B. papyrifera* leaf extract was more obvious in testes and uteruses than in ovaries. The testis coefficients of the CK were higher than those of all treatment groups after 1 day and 20 days (Figure 3A); notably, the testis coefficient of group IV was extremely lower than that of the CK after 1 day (*p* < 0.05). After 10 days, the testis coefficients of groups I, IV, and V increased and were higher than that of the CK; a significant difference (*p* < 0.05) was found between the CK and group II, and another between the CK and group IV.

The uterus coefficients of some groups were insignificantly higher than that of the CK (*p* > 0.05), while that of group IV was significant lower compared with that of the CK (*p* < 0.05). Evidently, the coefficient of the CK was superior both after 1 day and 20 days (Figure 3B). Nonetheless, the female ovary coefficients increased after the extract treatment (Figure 3C). The ovary coefficients of the CK were inferior to those of treatment groups I, III, IV, and V (*p* < 0.05), especially after treatment for 1 day, but the inferiority of the CK weakened over time.

Similarly with the organ coefficients, the physical development of the testes and uteruses was inferior after treatment. The testis volume of the CK was greater after 1 day and 20 days, while it was less than that of most of the treatment groups after 10 days (Figure 3D). The testis volume of group III was significantly lower than that of the CK after 1 day (*p* < 0.05), and that of group IV was significantly higher than that of the CK after treatment for 10 days (*p* < 0.05). Other comparisons showed no significant differences (*p* > 0.05). From the comparison of uterine lengths, the CK demonstrated superiority over time (Figure 3E). However, the CK showed significant inferiority after 1 day (Figure 3B) when its length was the shortest, and it was significantly weaker than that of group III (*p* < 0.05) and groups IV and IV (*p* < 0.01). This changed after 10 days, when only the uterine length of group I was significantly longer than that of the CK (*p* < 0.01), and the other groups measured less than the CK. Even group IV measured significantly shorter than the CK (*p* < 0.05), and group III and V were extremely and significantly shorter than the CK (*p* < 0.01). There was no statistically significant difference found after 20 days; however, the uterine length of the CK was higher than that of all treatment groups.

### 3.3. Effect of Extract on Sex Hormones

The *B. papyrifera* leaf extract showed positive effects on the measured sex hormones except for male testosterone (Figure 4). The testosterone levels of group I maintained slight superiority with no significance (*p* > 0.05) compared with those of the CK during the experiment, while the other four groups exhibited generally lower levels than those of the CK. (Figure 4A). Though the levels of groups III and V increased and were higher than those of the CK to a certain degree after 10 days, the levels of group V were significantly greater than those of the CK (*p* < 0.05), but then they decreased after treatment for 20 days. It is obvious that the testosterone levels of the CK were superior, especially compared with those of groups II and IV.

LH levels were measured both in males and females, which demonstrated that the stimulative effects were less inhibitive over time (Figure 4B,E). The LH levels fluctuated with changes in dose. After treatment for 1 day and 20 days, the male LH levels of the CK were still higher than those of groups II and IV, showing significant superiority. The LH levels of the CK decreased after 20 days, when those of the treatment groups demonstrated superiority. Unlike in males, the female LH levels of the CK were lowest from day 1, and most of the treated groups exhibited significantly superior levels throughout the experiment, compared to those of the CK.

Compared with the females in the CK, the sex hormone levels of the treated groups demonstrated remarkable superiority; as a whole, the extract enhanced the levels of E2 and FSH. The E2 levels of the CK were lower than those of the treated groups from the beginning (Figure 4C), and their levels were even lower after 20 days, though they seemingly improved from 1 day to 10 days. The benefit of the extract on FSH levels was not as significant as that on the above two female hormones (Figure 4D). The variations were extraordinary in groups IV and, in which the voles were treated with higher extract content. The FSH levels of groups IV and V were similar and increased throughout the experiment, even though they were significantly lower than those of the CK after 1 day (*p* < 0.05). Remarkably, their FSH levels exceeded those of the CK after treatment for 10 days and were significantly superior after 20 days (*p* < 0.05).

### 3.4. Variation In Male Sperm Parameters

There was no significant variation (*p* > 0.05) in the sperm quantity of the treated groups compared with that of the CK (Figure 5A). The CK group fluctuated widely, with a large standard error, meaning that it did not demonstrate significant superiority even though its mean quantities were higher than those of most treated groups after 1 day and 20 days. Nonetheless, the extract can be considered to have no positive effect on the sperm quantity of male voles.

The results show that the extract could reduce the activity of sperm over time (Figure 5B). After treatment for 1 day, the activities of most groups were higher than those of the CK except for group IV, but no significant differences were observed (*p* > 0.05). The activities of groups II, III, and IV obviously decreased, while those of groups I and V increased slightly after 10 days, and those of the CK were significantly higher than those of groups II and III (*p* < 0.05). After 20 days, the sperm activities of all the treated groups were lower than those of the CK, indicating the CK’s superiority, which remained significant compared with the sperm activities of groups II and III (*p* < 0.05).

Based on abnormal sperm, the extract demonstrated a teratogenic effect on the male *M. fortis* (Table 1, Figure 6). The types of abnormal sperm are shown in Figure 6. Though there were a few significant differences among the CK and treatment groups, most of the abnormal proportions in the treatment groups were higher than those of the CK. After 1 day, the abnormal proportion of the CK was 16.59% ± 0.87%, and only that of group IV was a bit lower; those of the other groups were all higher, and the abnormal proportion of group I was 23.55% ± 0.48%, which was significantly higher than that of the CK (*p* < 0.01). However, the abnormal proportions of groups I and IV fluctuated largely after being treated for 10 days; group I was the only group with an abnormal proportion lower than that of the CK. At this time, the abnormal proportion of the CK was higher than those of four groups and showed significant differences with groups III (*p* < 0.05) and IV (*p* < 0.01). After 20 days, the abnormal proportion of the CK was lower than 10%, while the others were higher; meanwhile, group IV still showed significant inferiority compared with the CK (*p* < 0.01).

### 3.5. The Reproductive Characteristics of Coupled Voles

Not many reproductive events were recorded. However, according to the observed phenomena, the inhibition of the *Broussonetia papyrifera* extract on the voles could be inferred. The reproduction parameters of the CK were greater than those of most groups (Table 2). Though the reproductive time of the CK was a bit longer than for groups IV and V, the CK produced a greater number of fetuses than the other two groups, although their weights were close. The differences between the CK and groups I, II, and III were more obvious. Group III was the only one superior to the CK in reproduction, with a lower reproductive time and a larger weight of fetuses. Although close in reproductive time and with the same fetus count, group I showed obvious inferiority in terms of fetal weight, which was only 2.900 ± 0.091 g, which is much lower than that of the CK (3.900 ± 0.102 g). Moreover, although the fetal weight of group II was 3.922 ± 0.083, which is a bit larger than that of the CK, its fetal count was lower. Finally, the main inferiority of group II was that its coupled voles required a much longer reproductive time compared with that for the CK and other groups.

## 4. Discussion

The present study attempts to reveal the antifertility effect of methanol extract from *B. papyrifera* leaves on *M. fortis*. This study also evaluated the potential effects on other physiological parameters and found that after treatment with the extract, the body weight of the voles remained unchanged and the growth of the heart, liver, lungs, and kidneys was also minimally influenced. These results show that the *B. papyrifera* leaf extract has hardly any harmful effects on the growth of *M. fortis* since the metabolic processes of these organs were normal. Overall, the comparisons of the reproductive parameters of the groups suggest that *B. papyrifera* leaf extract could potentially inhibit the growth of the reproductive organs of *M. fortis* and exert efficient teratogenicity on male sperm. Studies have previously revealed that plant extracts can inhibit the fertility of males, such as by reducing sperm motility and testosterone levels and increasing sperm deformity [54,55]. These results are similar to those for male voles in this study. As a whole, the fertility of female voles did not suffer obvious inhibition. Steroidal hormones, important for the development of sexual organs, demonstrated a close correlation. The development of female ovaries improved after treatment, and this may be because of the higher hormone levels. However, there is no certain evidence proving a positive correlation between sexual hormones and reproduction, as mating behavior can occur with very low levels of sexual hormones [56]. Based on the inferior reproduction of the treated coupled voles, some effects may have gone undetected in our experiment, such as estrous cycles, mating behavior, and pregnancy rate; these may also be attributed to the male voles.

In the 21st century, fertility control has become a humane, sustainable technology with few negative effects for controlling rodent damage and thus has attracted substantial attention. Scholars have proposed to control rodent damage by developing sexual sterilants for males [29]. Based on the rodent’s robust multifactor reproductive strategy, fertility control may be an efficient measure for controlling the overpopulation of most kinds of rodents with fewer expenditures. From this viewpoint, *B. papyrifera* leaves can at least be used to induce antifertility on male rodents. Although developing an antifertility agent from a plant source is cost-effective with low toxicity, in practical application, high costs and production difficulties will still be encountered. In China, sterilants from plant sources that have been applied for the fertility control of pest rodents include *Tripterygium wilfordii*, gossypol, *Ruta graveolens*, *Camellia oleifera*, radix trichosanthis (*Trichosanthes kirilowi*), colchicine, semen ricini (*Ricinus communis*), aeruginous turmeric rhizome (*Curcumae aeruginosae*), and neem (*Melia azederach*). Though all of these were developed as baits for animal experiments in the laboratory or sprayed in the wild for animal foraging, with certain effects, their practical applications have been unsatisfactory [57]. There are a number of elements that should be considered so these active plant substances remain effective in the wild, such as costs and inertia influenced by weather. Global studies have conducted many tests on the safety and efficacy of natural substances in the lab, but problems still arise during real-world use [58,59]. Thus, the application of fertility control has remained stagnant. Compared with these plants, the advantages of *B. papyrifera* leaves might be obvious. First, *B. papyrifera* is widely planted and naturally grow in China [44,60], which means that the leaves are abundant and available; even people who usually suffer from rodent pests could collect the leaf materials locally, greatly saving on the cost of plant material collection and transportation. Second, the crude extract of *B. papyrifera* leaves already has an efficient inhibitive effect on *M. fortis* according to this research. This means that this plant resource is more economical in the production process. Third, these voles are strongly attracted to *B. papyrifera* leaves for feeding. Though fertility control has greater potential than conventional rodenticides for controlling rodents [61,62,63], it is undeniable that its advantages and disadvantages are relative; as some scholars have proposed, the effects of sterilants are not immediate because they do not directly cause rodent mortality [34]. Although people would rather kill all rodents in a small environment, it may be more desirable to control their population in the ecosystem [64]. In this case, neither sterilants nor rodenticides are excellent measures against rodent pests. However, their combined use may be applicable and efficient in some cases. For example, in a rodent population outbreak, rodenticides could first be used to eliminate most rodent individuals, and then sterilants could be used to render the remaining rodents sterile.

Excavating active substances is also significant for the evolution of plant sterilants. The compounds and substances in the methanol extract of *B. papyrifera* leaves were determined via metabolomic analysis. A total of 5170 metabolites were detected in the methanol extract of *B. papyrifera* leaves and classified into 17 categories (Figure A2). Studies have demonstrated that plants rich in secondary metabolites such as flavonoids, terpenes, and alkaloids show antifertility activity in animals, including impeding ovulation, impeding spermatogenesis, reducing sperm activity and count, and disturbing the breeding cycle [65,66]. These kinds of secondary metabolites have been identified in the methanol extract. Studies have also revealed that plant extracts also contain toxic compounds like glycosides, anthraquinones, tannins, organic acids, and toxic minerals, which have a toxic effect on the liver and testes [67]. Most of these metabolites were also found in our study; thus, it is indeed important to fully understand the effectiveness of the substance due to the large number of active compounds in the methanol extract of *B. papyrifera* leaves. In addition, the dose and treatment pattern can cause variations in the effects [68]. In this study, when the dose was gradually increased and reached a particular degree, its effect tendency changed to the opposite. There were different effects on voles between single treatments and repeated treatments; for example, the intake of group IV after being treated for 10 days was the same as for group V after 1 day. However, the effect showed some differences: the voles responded more effectively with the single treatment in the early stage of the experiment; otherwise, they should be treated long term, for at least more than 20 days according to our results. The treated groups can also be regarded as 25 groups because the frequency of gavage with the same dose per day was different; this may be related to the metabolism, bioavailability, and blood concentration of these efficient substances, indicating the intricacy of their mechanism. Therefore, the application of sterilants should be mindful of doses and measures, and further exploration should be carried out to determine the relationship.

## 5. Conclusions

This study reports the potential of *B. papyrifera* leaves as an economic and effective sterilant for *M. fortis*. A crude methanol extract showed potent inhibition, especially on male fertility, impeding the growth of sex organs and reducing sperm quality. This study supports that *B. papyrifera* leaves would be applicable in a male contraceptive agent. The example of *B. papyrifera* leaves also exemplifies a new idea for developing rodent sterilants as it supports stagnant application with effectiveness and low cost. A series of lab experiments and field verifications are required to reveal the effective mechanisms and practicability of *B. papyrifera* leaf sterilants. Moreover, the opportunistic discovery of plants with antifertility activity should be encouraged and supported to further explore their feasibility as pest chemosterilants and enrich the communication and practices of pest management.

## Figures and Tables

**Figure 1 biology-14-00056-f001:**
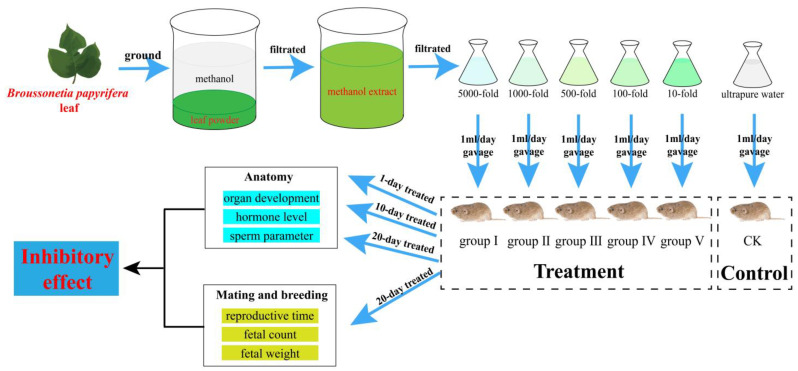
The treatment framework and methods in this study.

**Figure 2 biology-14-00056-f002:**
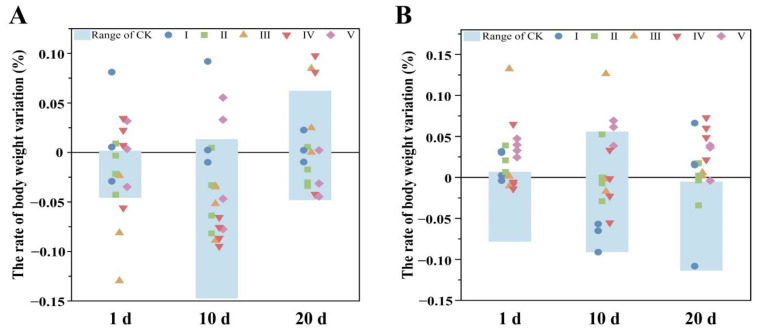
The variation in body weight of voles in different groups ((**A**) male, (**B**) female). The range of the CK was regarded as the standard; samples in each treated group were compared with the CK, showing that discrepant proportions were small.

**Figure 3 biology-14-00056-f003:**
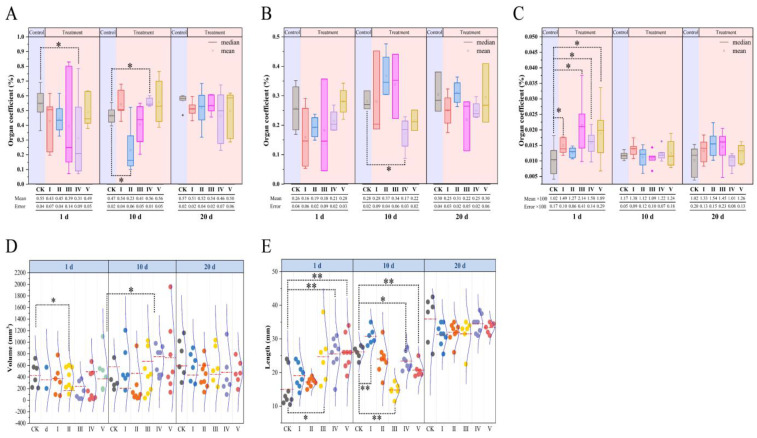
Characteristics of sex organs in different groups ((**A**) showing male testis coefficient of treated groups being lower than that of CK, (**B**) showing female uterus coefficient of treated groups compared with that of CK, (**C**) showing female ovary coefficient of treated group being slightly higher than that of CK, (**D**) showing male testis volume of CK being greater than that of treated groups, and (**E**) showing female uterus length of treated groups being larger than that of CK). * represents *p* < 0.05; ** represents *p* < 0.01.

**Figure 4 biology-14-00056-f004:**
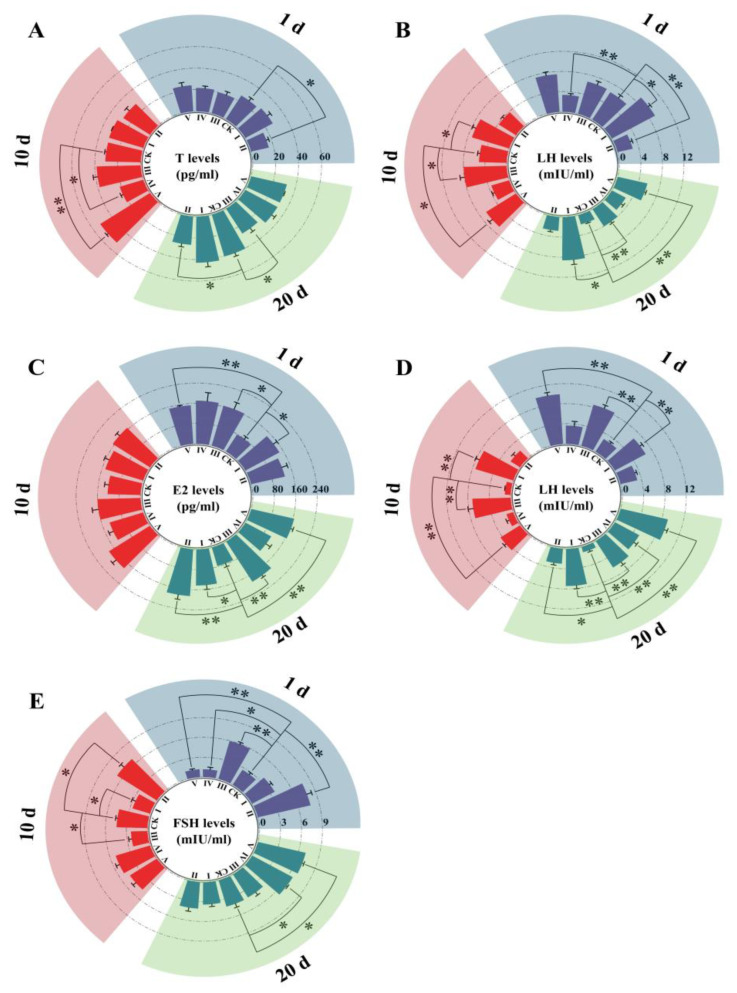
The levels of serum sex hormone in the groups: (**A**) male testosterone, (**B**) male luteinizing hormone (LH), (**C**) female estradiol (E2), (**D**) female luteinizing hormone (LH), (**E**) female follicle-stimulating hormone (FSH). * represents *p* < 0.05; ** represents *p* < 0.01.

**Figure 5 biology-14-00056-f005:**
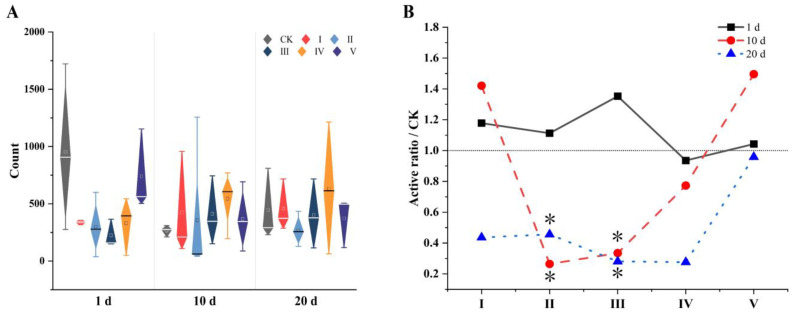
The comparison of male sperm parameters ((**A**) showing sperm count of treated individuals with no advantage compared with CK, (**B**) showing sperm activity of most treated individuals being lower than that of CK). * represents *p* < 0.05 compared with CK.

**Figure 6 biology-14-00056-f006:**
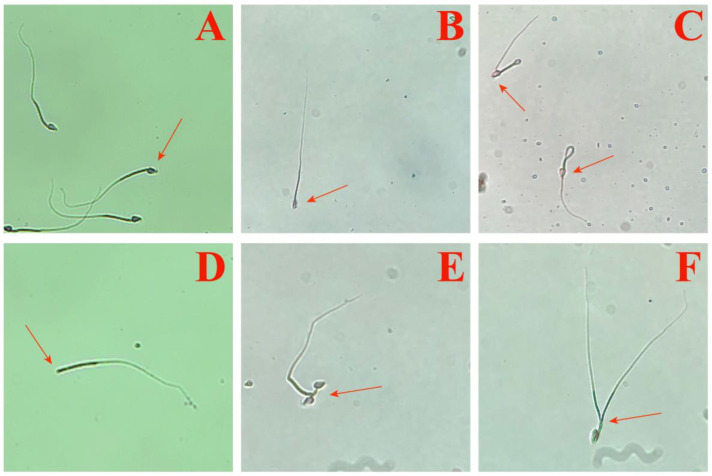
Sperm morphology ((**A**) normal sperm, (**B**) amorphous, (**C**) coiled tail, (**D**) no head, (**E**) two head, (**F**) two tails).

**Table 1 biology-14-00056-t001:** The proportion of abnormal sperm and comparison between treatment groups and CK.

Group	Abnormal Proportion (Mean ± SE %, *p* Value = Group/CK)		
	1 d	10 d	20 d
CK	16.59 ± 0.87	9.77 ±1.05	8.29 ± 1.94
I	23.55 ± 0.48 **	9.15 ±0.92	10.45 ± 3.94
II	17.69 ± 1.95	15.45 ± 5.51	18.17 ± 3.16
III	25.15 ± 5.33	19.01 ±2.10 *	17.54 ± 3.47
IV	16.15 ± 2.03	22.79 ± 1.32 **	25.71 ± 2.94 **
V	18.42 ± 1.13	11.42 ± 2.76	12.91 ± 0.98

Note: * represents *p* < 0.05 and ** represents *p* < 0.01 compared with CK.

**Table 2 biology-14-00056-t002:** Characteristics of reproduction in different groups.

Group	Reproductive Time (d)	Number of Fetuses	Weight of Fetuses (g)
CK	24	4	3.900 ± 0.102
I	22	4	2.900 ± 0.091
II	28.333	3	3.922 ± 0.083
III	21.667	4	4.117 ± 0.139
IV	22.667	3.333	3.990 ± 0.129
V	22.5	3	3.967 ± 0.080

## Data Availability

The data are contained within this article.

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
