# Peer review of "Broussonetia papyrifera Extract Can Be Used as a Raw Material Source for a Sterility Agent for Microtus fortis"

_biology, 2025, doi:10.3390/biology14010056_

Round 1
Reviewer 1 Report
Comments and Suggestions for Authors
General impression
This is an interesting scientific document that contributes to our knowledge about new possibilities of using plant extract, now Broussonetia papyrifera extract as a tool for fertility control of voles.
Generally, the fertility control of different rodent species is under development, and this paper will can initiate a whole series of subsequent researches.
The study is adequately designed, the paper has a good structure, the goals are clearly defined and the project was analyzed and presented.
My impression of this study is positive and I think that this project will be able to play an important role in international level.
I recommend acceptance of this manuscript with the minor revision.
For the first, the language should be improved. Also, many type errors through the Mns should be fixed.
References not checked.
Following some questions and suggestions:
General comments
P2 Line 55-57: the cited literature provides data on rats that are not the subject of research. I would ask the authors to focus more on voles, their maintenance and control measures. The text often mentions "rat", and refers to other types of rodents, such as field rodents (mice, voles).
P2 Line 57: ... which are quick action… ? Anticoagulants are not fast-acting rodenticides. Please, check it. The same comment on Line 97.
P2 Line 59. The rodent resistance development. I'm not sure that statement is entirely true. Please check it. There are numerous works that indicate the methods and reasons for the development of resistant populations.
P2 Line 63-65: “This consequence has led, not 63 only to potentially harmful on rodent predators and environment [16, 17], but also to the 64 evolution of resistance with rodents to these chemical rodenticides”. Please, check this sentence.
P2 Line 90: …surrounding village … The voles migrate into houses?
P3 Line 101: It is very interesting that an animal consumes food that limits its reproductive abilities. Do you have information that this is repeated somewhere else, the relationship between other animals and plants?
Has the natural influence of the plant Broussonetia papyrifera on vole numbers been observed in that region?
P7 Line 352: Do you have data on the use of sterilants outside of China? What is happening in Europe, USA, Australia?
Suggestion
- I would like to see a more information about active substance which is farmers used against voles population (M. fortis). Also, the information about ingredients carriers. The efficacy of a rodenticide bait may be dependent on the quality of the bait base rather than on the active ingredient. There is a plenty references about it as a:
DELGADO GARCÍA, J.D.. "Selection and treatment of fleshy fruits by the Ship Rat (Rattus rattus L.) in the Canarian laurel forest" Mammalia, vol. 64, no. 1, 2000, pp. 11-18. https://doi.org/10.1515/mamm.2000.64.1.11
Robards G. E. Saunders Glen (1998) Food preferences of house mice (Mus domesticus) and their implications for control strategies. Wildlife Research 25, 595-601. https://doi.org/10.1071/WR97109
Jokić G., Vukša M., Elezović I., Đedović S., Kataranovski D. (2012): Application of grain baits to control common vole Microtus arvalis (Pallas, 1778) in alfalfa crops, Serbia. rchives of Biological Sciences 2012 Volume 64, Issue 2, Pages: 629-637
https://doi.org/10.2298/ABS1202629J
- If you have the opportunity, wherever possible, because of the experimental design you followed in the work, refer to some official method or methodology that has already been published in some books or peer-reviewed papers.
Author Response
Dear Reviewer,
We feel great thanks for your professional review work on our manuscript. We have studied comments carefully and have made correction which we hope meet with approval. As you are concerned, there are several problems that need to be addressed. According to your comments, we have made corrections to our manuscript, and with tracking pattern and marked in blue. The main correction and response to your comments are as flowing.
Comment 1
P2 Line 55-57: the cited literature provides data on rats that are not the subject of research. I would ask the authors to focus more on voles, their maintenance and control measures. The text often mentions "rat", and refers to other types of rodents, such as field rodents (mice, voles).
Response 1
Thank to you for your careful check, accroding to your suggestion, we have changed some references focus on voles, and deleted the original references.
Comment 2
P2 Line 57: ... which are quick action… ? Anticoagulants are not fast-acting rodenticides. Please, check it. The same comment on Line 97.
Response 2
We sincerely thank you for pointing out the mistake, in revised manuscript, we have amended the sentences refer to added reference.
Comment 3
P2 Line 59. The rodent resistance development. I'm not sure that statement is entirely true. Please check it. There are numerous works that indicate the methods and reasons for the development of resistant populations.
Response 3
Thanks for your scientifically comment, we checked the sentence and its meaning, and we supplemented some reference may support the statement.
Comment 4
P2 Line 63-65: “This consequence has led, not 63 only to potentially harmful on rodent predators and environment [16, 17], but also to the 64 evolution of resistance with rodents to these chemical rodenticides”. Please, check this sentence.
Response 4
According to your comment, we changed the description, because rodents migrate, they may ingest little rodenticides substances, that may not kill they, but activate their immune, cause the further resistance. We would like the new descriptions could be clearly.
Comment 5
P2 Line 90: …surrounding village … The voles migrate into houses?
Response 5
Thank for your comment, as you surmise, people living along the edge of the Dongting Lake, they farm with the benefit of the lake water, so when water level rise, the voles have to migrate out of the wetland, they intrude the space of residents near the lake, and cause the crop damage and sanitary event. These were the mainly pest. To clear the information, we added one more sentence in the revised manuscript.
Comment 6
P3 Line 101: It is very interesting that an animal consumes food that limits its reproductive abilities. Do you have information that this is repeated somewhere else, the relationship between other animals and plants?
Response 6
Thanks for your valueble feedback, this phenomenon is the origin that we started studing the relationship between Broussonetia papyrifera and Microtus fortis. According to your comment, we have viewed some references, it is a pity that we have not find another one relationship like this, we still focus on the phenomenon when we read other references. There are some food showed antifertility activity to animals, such as Avicennia Marina, Daucus carota, but not their edible part, or they are processed.
[1] Rodiani D, Maryono T, Ramdini D A. Avicennia marina: a natural resource for male anti-fertility in family planning[J]. International Journal of Design & Nature and Ecodynamics, 2023, 18(5): 1077-1085.
[2] Ganguly M, Hazarika J, Sarma S, et al. Estrogen receptor modulation of some polyphenols extracted from Daucus carota as a probable mechanism for antifertility effect: An in silico study[J]. Journal of Theoretical and Computational Chemistry, 2020, 19(06): 2041004.
Comment 7
Has the natural influence of the plant Broussonetia papyrifera on vole numbers been observed in that region?
Response 7
We sincerely thanks to you for your valuable feedback, we discover the feeding relationship between Broussonetia papyrifera and Microtus fortis in the wild showed in pictures, it interested us to start these explorations. Our stories still around the laboratory, we think the rodent plant sterility agent should be well palatability, and can inhibited the reproductive ability. According to our previous study, we think Broussonetia papyrifera leaf satisfied above two characters. Then we would like to excavate the key substance in the leaf. The viewpoint you remind us also the further important work to do to reveal the mechanism in the relation between Broussonetia papyrifera and rodents.
Comment 8
P7 Line 352: Do you have data on the use of sterilants outside of China? What is happening in Europe, USA, Australia?
Response 8
Thanks to you for your rigorous comment, we have read many references, and supplemented some in our manuscript. Lots of natural production has the antifertility effect on rodents, but most are verified in lab. We have supplemented some discussion in manuscript. As you worried, the utilization outside, is a scientific further work need to be verified.

Reviewer 2 Report
Comments and Suggestions for Authors
Microtus fortis is an important agricultural pest species in China. Broussonetia papyrifera is a widely distributed, fast-growing shrub plant species, and is often used as animal feed. In this study, the methanol extract of leaves of B. papyrifera were used to fed M. fortis adults, and the morphological index of organs, hormone, sperms of the adults were measured and compared between treatment and control. The results showed that the growth of sex organs of M. fortis were inhibited, and the testosterone level, sperm quality and quantity of males were reduced after feeding by the extract. These findings indicate that M. fortis might contain plant sterilant substance, and can be used for biological control of rodent pests. The result also provide insights into the utilization of B. papyrifera. I suggest accept after revise.
1. Materials and Methods
1.1Are the 180 mature M. fortis all the same age? Please add the age detail.
1.2 The treatments are not descripted adequately. Does 1-day treated, 10-day treated, 20-day treated mean fed the rodents 1day, 10days, 20 days respectively? If so, there should have 15 treatments, because there are 5 concentrations of the extract.
1.3 When were the data collected from the rodents after treatment? All after 20 days?
2. Results
2.1 Footnote all tables and figures.
2.2 All abbreviations should be full presented when they first appeared.
2.3 Figure 4, D should be “level of female follicle-stimulating hormone”, not “HL levels” (same as B).
2.4 Table 2, Are the data of “Reproductive time(day)” and “Count of fetuses” means of the replicates? How many replicates (rodent individuals)?
3. Use “d” instead of “days” in all figures and tables.
4. Broussonetia papyrifera should be listed in the Key words.
Comments on the Quality of English LanguageCheck the language carefully.
L39, “also” is not proper in this sentence, should be deleted.
Some words are not properly used. Such as L40, “weight were worse”, “were worse” should be “was reduced”.
Some sentences are too long, with different subjects. They should be separated to 2 or more sentences with same subject.
L52, “Some of they” should be “Some of them”
L109-L111, there is no subject in the sentence.
Author Response
Dear Reviewer,
On behalf of all the contributing authors, I would like to express our sincere appreciations of your constructive comments concerning our article. These comments are all valuable and helpful for improving our manuscript. We have tried our best to make all the revisions clear according to your suggestion, and using tracking pattern and marked in blue. There are some language problems in the text, we also modified the text with tracking pattern, we hope it would be more correct to read. If the problem still exists, we would consider to polished the final manuscript. The response to each comment is as flowing:
Comment 1
- Materials and Methods
1.1 Are the 180 mature M. fortis all the same age? Please add the age detail.
Response 1
Thanks for your valuable comment, the animals were approximately 8 weeks old, so they have the normal reproductive ability. We have added the detail at the sentence in the revised manuscript. (Line 132
Comment 2
1.2 The treatments are not descripted adequately. Does 1-day treated, 10-day treated, 20-day treated mean fed the rodents 1day, 10days, 20 days respectively? If so, there should have 15 treatments, because there are 5 concentrations of the extract.
Response 2
We think this is an excellent comment, as you mentioned, the treatments were much more than 5 groups, in fact, we focus on the different between treated doses and the blank control (CK). However, there are more relationship among there doses, which also be emphasized from you. For example, one treatment of Group Ⅴ dose, the dosage is equal with ten treatment of Group Ⅳ, the same situation also existed among other groups, it demanded the deeply analysis focus on pharmacology, such as bioavailability and blood concentration of the active substance. We have added related discussion in the revised manuscript, these also suggested us, a lot of work should be carried out in future to explore the response of rodent to Broussonetia papyrifera.
Comment 3
1.3 When were the data collected from the rodents after treatment? All after 20 days?
Response 3
We sincerely thank you for your carefully review, to clear the information, we added the detail in the corresponding location of text, and explain at the caption of figure 1.
Comment 4
- Results
2.1 Footnote all tables and figures.
Response 4
According to your suggestion, we have added relevant footnote in tables and figures, except a few which showed the direct data.
Comment 5
2.2 All abbreviations should be full presented when they first appeared.
Response 5
Thank to you for your reminding, we have checked the manuscript, and adjusted the first appearance of subsequent abbreviations
Comment 6
2.3 Figure 4, D should be “level of female follicle-stimulating hormone”, not “HL levels” (same as B).
Response 6
Thank you for pointing out the mistake, we have checked the data and figure, the problem occurred in the figure caption, in the revised manuscript, we corrected it. A and B were the parameter of male, C, D and E were female. Levels of LH were both measured in male and femal.
Comment 7
2.4 Table 2, Are the data of “Reproductive time(day)” and “Count of fetuses” means of the replicates? How many replicates (rodent individuals)?
Response 7
Thanks for your comment. There are three couple voles in each group, it is insufficient that we have not obtain more than three reproductions in each group, but we would like to show the readers the comparative phenomenon between CK and treated groups, to discuss the response of reproduction of voles to the Broussonetia papyrifera extract. “Reproductive time (day)” were the mean time of reproductive event. Then “Quatiny of fetuses” was used instead of “Count of fetuses”, said that the mean number of the newborns. In the manuscript, we also try our best to explain this situation.
Comment 8
Use “d” instead of “days” in all figures and tables.
Response 8
As suggested by you, we have used “d” instead of “days” in figures and tables.
Comment 9
Broussonetia papyrifera should be listed in the Key words.
Response 9
Thanks for your valuable comment, inspired by you, we added “paper mulberry” in Key word, which is the vulgo of Broussonetia papyrifera, considered that the title of manuscript include “Broussonetia papyrifera”.
Comment 10
L39, “also” is not proper in this sentence, should be deleted.
Response 10
As your suggestion, we have deleted the word in the sentence.
Comment 11
Some words are not properly used. Such as L40, “weight were worse”, “were worse” should be “was reduced”.
Response 11
Thanks for your suggestion. We have checked the similar problem, and polished the whole manuscript.
Comment 12
Some sentences are too long, with different subjects. They should be separated to 2 or more sentences with same subject.
Response 12
As you said, the language should be improved, the revised manuscript have been professionally polished to enhance the expression.
Comment 13
L52, “Some of they” should be “Some of them”
Response 13
We are sorry for the mistake, and it has been amended in the revised manuscript.
Comment 14
L109-L111, there is no subject in the sentence.
Response 14
Thank you for your comment, we have modified this sentence in the revised manuscript.
